# Acetylcholinesterase Inhibitory Activities of Essential Oils from Vietnamese Traditional Medicinal Plants

**DOI:** 10.3390/molecules27207092

**Published:** 2022-10-20

**Authors:** Nguyen Huy Hung, Pham Minh Quan, Prabodh Satyal, Do Ngoc Dai, Vo Van Hoa, Ngo Gia Huy, Le Duc Giang, Nguyen Thi Ha, Le Thi Huong, Vu Thi Hien, William N. Setzer

**Affiliations:** 1Center for Advanced Chemistry, Institute of Research and Development, Duy Tan University, 03 Quang Trung, Da Nang 550000, Vietnam; 2Department of Pharmacy, Duy Tan University, 03 Quang Trung, Da Nang 550000, Vietnam; 3Graduate University of Science and Technology, Vietnam Academy of Science and Technology, 18-Hoang Quoc Viet, Cau Giay, Hanoi 10000, Vietnam; 4Institute of Natural Products Chemistry, Vietnam Academy of Science and Technology, Hanoi 100000, Vietnam; 5Aromatic Plant Research Center, 230 N 1200 E, Suite 100, Lehi, UT 84043, USA; 6Faculty of Agriculture, Forestry and Fishery, Nghe An College of Economics, 51-Ly Tu Trong, Vinh City 43000, Vietnam; 7School of Natural Science Education, Vinh University, 182 Le Duan, Vinh City 43000, Vietnam; 8Drug, Comestic and Food Quality Control Center of Ha Tinh Province.46, Ha Hoang Street, Thach Trung Commune, Ha Tinh City 481300, Vietnam; 9Faculty of Hydrometeorology, Ho Chi Minh City University of Natural Resources and Environment, Ho Chi Minh City 70000, Vietnam; 10Department of Chemistry, University of Alabama in Huntsville, Huntsville, AL 35899, USA

**Keywords:** Alzheimer’s, atractylone, *trans*-carveol, Lamiaceae, pesticide

## Abstract

Essential oils are promising as environmentally friendly and safe sources of pesticides for human use. Furthermore, they are also of interest as aromatherapeutic agents in the treatment of Alzheimer’s disease, and inhibition of the enzyme acetylcholinesterase (AChE) has been evaluated as an important mechanism. The essential oils of some species in the genera *Callicarpa*, *Premna*, *Vitex* and *Karomia* of the family Lamiaceae were evaluated for inhibition of electric eel AChE using the Ellman method. The essential oils of *Callicarpa candicans* showed promising activity, with IC_50_ values between 45.67 and 58.38 μg/mL. The essential oils of *Callicarpa sinuata, Callicarpa petelotii, Callicarpa nudiflora, Callicarpa erioclona* and *Vitex ajugifolia* showed good activity with IC_50_ values between 28.71 and 54.69 μg/mL. The essential oils *Vitex trifolia* subsp. *trifolia* and *Callicarpa rubella* showed modest activity, with IC_50_ values of 81.34 and 89.38, respectively. *trans*-Carveol showed an IC_50_ value of 102.88 µg/mL. Molecular docking and molecular dynamics simulation were performed on the major components of the studied essential oils to investigate the possible mechanisms of action of potential inhibitors. The results obtained suggest that these essential oils may be used to control mosquito vectors that transmit pathogenic viruses or to support the treatment of Alzheimer’s disease.

## 1. Introduction

Acetylcholinesterase (AChE) is a cholinergic enzyme primarily found at postsynaptic neuromuscular junctions, especially in muscles and nerves. It immediately breaks down or hydrolyzes acetylcholine (ACh), a naturally occurring neurotransmitter, into acetic acid and choline. The primary role of AChE is to terminate neuronal transmission and signaling between synapses to prevent ACh dispersal and activation of nearby receptors [1]. Hydrolysis of acetylcholine is required to allow a cholinergic neuron to return to its resting state after activation [2]. 

When AChE is inactivated, e.g., by an organophosphorus or carbamate ester, the enzyme is no longer able to hydrolyze ACh; the concentration of ACh in the junction remains high, and continuous stimulation of the muscle or nerve fiber occurs, resulting eventually in exhaustion and tetany [3]. Inhibition of AChE leads to excess synaptic acetylcholine levels, over-stimulation of cholinergic receptors, alteration of postsynaptic cell function and consequent signs of cholinergic toxicity [4]. This inhibition leads to an accumulation of acetylcholine in the synapses, which in turn leaves the acetylcholine receptors permanently open, resulting in the death of the organism (e.g., insecticidal activity) [5].

A decrease in acetylcholine levels in synaptic clefts is thought to be responsible for Alzheimer’s disease [6]. Cholinergic deficiency is an early and consistent presentation in Alzheimer’s patients [7]. Although the role of AChE in Alzheimer’s disease is unclear, it is by far the most viable therapeutic target for improving symptoms of the disease [8].

According to the World Health Organization (WHO), Alzheimer’s disease accounts for 50 to 60 percent of all dementia cases [9]. Currently, there are disadvantages associated with current Alzheimer’s disease chemotherapeutics. For example, tacrine was observed to have serious side effects such as elevated liver transaminases and gastrointestinal problems [10,11]; donepezil has been reported to exhibit toxicity similar to beta-blocker overdose and colitis [12]; the toxicity of rivastigmine is thought to be similar to that of other carbamates and organophosphates with features of muscarinic and nicotinic stimulation [13]. Essential oils and their chemical constituents have been shown to have effects on the central nervous system, including in the treatment of Alzheimer’s disease and Parkinson’s disease [14]. The components of essential oils are characterized by their small size and lipophilicity, thus facilitating movement across the blood–brain barrier [15,16]. Their characteristic volatility may facilitate their use in inhalation, avoiding the metabolic channel, with its facilitator denaturing the active components [17]. There have been some clinical reports that aromatherapy improves memory and alleviates psychobehavioural symptoms in Alzheimer’s patients [18,19,20,21,22]. In vivo mouse models have also shown beneficial effects of essential oils for the prevention and treatment of Alzheimer’s disease [23]. 

In Vietnamese folk medicine, *Callicarpa candicans* and *Callicarpa longifolia* are used as tonics for women after childbirth [24,25]. *Callicarpa rubella* is used as a medicine for treating scabies, rheumatism, and contusions [24]. *Callicarpa nudiflora* and *Callicarpa macrophylla* are used as medicines to treat stomach bleeding, nosebleeds, fire burns, hepatitis, and contusions [24]. *Callicarpa erioclona* is used as a remedy for gastrointestinal bleeding, for gonorrhea, as an insecticide, and as a poison for fish [24]. *Premna cambodiana* is used as a medicine for treating spermatorrhea [24]. *Vitex trifolia* subsp. *litoralis* and *Vitex trifolia* subsp. *trifolia* have been used in traditional Vietnamese medicine to relieve headaches, rheumatism, muscle pain, and neuralgia [24,26]. 

In this study, we investigated the AChE enzyme inhibitory activities of some essential oils of *Callicarpa, Premna*, and *Karomia* species from Vietnam and their main chemical constituents with the goal of finding essential oils as potential aromatherapy in the treatment of Alzheimer’s disease, as well as investigating potential sources of essential oils for controlling insect pest species.

## 2. Results and Discussion

### 2.1. Chemical Compositions of the Essential Oils

The main chemical compositions of the essential oils were reported in our previous studies, and are presented in Table 1, their structures are shown in Figure 1.

### 2.2. Acetylcholinesterase Inhibitory Activity

Acetylcholinesterase inhibitory activity of essential oils of *Callicarpa, Premna* and *Karomia* species from Vietnam and their major components have been presented in Table 2 and Table 3. The essential oils *C. candicans* (Da Nang), *C. candicans* (Quang Nam), *C. erioclona*, *C. sinuata*, *C. petelotii*, *C. nudiflora*, and *V. ajugifolia* showed good activity, with IC_50_ values between 28.71 ± 3.85 and 54.69 ± 3.05 μg/mL. The essential oils of *C. candicans* (Nghe An) and *P. corymbosa* exhibited strong activity, with IC_50_ values of 58.38 ± 2.95 and 73.35 ± 4.61 μg/mL, respectively. Two monoterpene hydrocarbon compounds, limonene and β-pinene, showed good activity, with IC_50_ values of 53.16 ± 4.08 and 71.45 ± 5.77 μg/mL, respectively. Meanwhile, caryophyllane compounds showed weak activity.

To the best of our knowledge, this is the first time *trans*-carveol has been evaluated for AChE inhibitory activity, and it exhibited an IC_50_ value of 676 ± 52 µM. Studying the AChE inhibition of monoterpenoids with a *p*-menthane skeleton observed that monoterpene ketones showed stronger inhibition than the alcohols, and the presence of an isopropenyl group improved the inhibitory strength [30]. Limonene, in the study by Miyazawa and co-authors, showed inhibition at a concentration of 1.2 mM in the range of 22.0–25.0%; however, our study recorded an IC_50_ value of 390.2 ± 30.0 µM. This difference may have been due to the experimental method and origin of the AChE used.

(*E*)-β-Caryophyllene has previously been reported to exhibit an inhibition of AChE from *Electrophorus electricus* (electric eel) at a concentration of 0.06 mM of 32% [31], while that of AChE from human erythrocytes gave an IC_50_ value of 147 ± 15 µM [32]. (*E*)-β-Caryophyllene oxide inhibited AChE of *E. electricus* at 200 µg/mL by 41.46 ± 2.66% [33], and at 250 µg/mL inhibited 35 ± 4.7% AChE of bovine erythrocytes [34]. β-Pinene inhibited the AChE of bovine erythrocytes with IC_50_ values around 1500 μM [32,34,35]. Myrtenal exhibited AChE inhibitory activity with an IC_50_ value of 0.17 mM [36]. α-Humulene exhibited weak inhibition of AChE with an IC_50_ value of 785.3 ± 66.0 μM.

Several studies have shown that essential oils with high concentrations of sesquiterpene derivatives exhibit moderate and weak AChE inhibitory activity. A study by Karakaya and co-authors clearly showed that an increase in the concentration of sesquiterpenoids reduced the AChE inhibition of the essential oil, at a concentration of 200 µg/mL, the essential oil from the aerial parts of *Salvia verticillata* subsp. *amasiaca* with 60.1% sesquiterpenoids inhibited 51.65 ± 2.05% while its floral essential oil with 78.9% sesquiterpenoids showed inhibition of 42.19 ± 1.55% [33]. Other studies support this trend, such as Salinas et al. 2020 [37], Ali et al. 2012 [38], and Siebert et al. 2015 [39]. Some of the results in this study that were consistent with this trend were those for *C. longifolia*, *C. macrophylla*, *P. cambodiana*, *V. pinnata, K. fragrans*, which had IC_50_ values between 144.3 and 221.85 μg/mL.

Some of the essential oils in this study, although characterized by absolute predominance of sesquiterpenoids, exhibited good and potent AChE inhibitory activity (see above). Similarly, the leaf essential oil of *Annona cherimola*, characterized by 73.87% of sesquiterpenoids including germacrene D (28.77%), bicyclogermacrene (11.12%), *E*-β-caryophyllene (10.52%), sabinene (9.05%), and β-pinene (7.93%), showed strong AChE inhibition with an IC_50_ value of 41.51 ± 1.02 µg/mL [40]. The essential oil of *Eugenia riedeliana* was characterized by an absolute predominance of sesquiterpenoids (94.2%), but also exhibited strong inhibition with an IC_50_ value of 67.3 μg/mL [41]. When increasing the content of sesquiterpenoids compared with monoterpenoids concentration, an increase in the ability of the essential oil to inhibit AChE was observed [42]. Studies by other groups such as da Silva Barbosa et al. [43] and Miyazawa et al. [44] have supported this trend. Research has shown that the main compound did not contribute to the activity of the essential oil containing it, and in such cases minor components were responsible [45]. Essential oils are a complex mixture of many compounds, most of which have not been studied for their AChE inhibitory activity or studies for their synergistic or antagonistic effects.

Furanosesquiterpenoids are a specific class of compound that has been reported to have AChE inhibitory activity in an in vivo model [46]. Commiterpenes A–C have been reported to have neuroprotective effects [47]. The leaf essential oil of *Eugenia uniflora* contains atractylone and 3-furanoeudesmene. Both the essential oil and a mixture of the two components showed antinociceptive activity [48]. The furan ring has been suggested to be involved in the inhibitory effect of AChE in previous studies [49]. The essential oil of *C. candicans*, rich in atractylone, showed excellent larvicidal activity against *Aedes aegypti* (LC_50_ = 2.7–5.34 μg/mL at 24 h) and *Culex quinquefasciatus* (LC_50_ = 1.20–2.04 μg/mL at 24 h) [27].

*Vitex trifolia* subsp. *litoralis* essential oil was characterized by monoterpenoid compounds accounting for 76%, including α-pinene (18.7%), sabinene (15.2%), 1,8-cineole (14.5%), and α-terpinyl acetate (12.7%). α-Pinene and 1,8-cineole have been shown to have synergistic effects [34,50]. α-Pinene exhibited inhibitory activity of AChE from bovine erythrocytes, with an IC_50_ value of 660 ± 40 μM [34]. Sabinene has been shown to exert AChE inhibition, with an IC_50_ value of 1296 ± 21 μM [51]. 1,8-Cineole exhibited an IC_50_ value of 6 μM for the inhibition of electric eel AChE [52], and 390 ± 60 μM for the inhibition of bovine erythrocyte AChE [34]. Although α-terpinyl acetate has not been reported to inhibit AChE, α-terpineol has nonetheless been reported with an IC_50_ value of 8400 ± 400 μM [53], and α-terpinene has been reported with an IC_50_ value of 1000 μM [30]. Essential oils asserted to be rich in monoterpenoid compounds in previous studies showed a tendency to exhibit good AChE inhibition. Essential oil of *Pinus nigra* subsp. *nigra* that included α-pinene (25.3%), limonene (22.6%), sabinene (12.8%), α-terpineol (8.3%), β-pinene (4.8%), and terpinolene (4.5%) showed an IC_50_ value of 94.4 ± 1.8 μg/mL [31].

*Vitex trifolia* subsp. *trifolia* demonstrated a strong inhibitory effect on AChE, with an IC_50_ value of 81.34 ± 3.57 μg/mL. Research by Liu et al. showed that sabinene has a synergistic effect with 1,8-cineole [45], while α-pinene and 1,8-cineole have also shown synergistic effects [34,50]. In addition, sabinene also exhibits a synergistic effect with other minor components such as limonene (1.2%) and linalool (0.1%) in the essential oil of *V. trifolia* subsp. *trifolia* [45]. *Vitex ajugifolia* showed good AChE inhibition with an IC_50_ value of 50.93± 4.81 μg/mL. α-Copaene has shown very strong synergistic effects in combination with both (*E*)-β-caryophyllene and α-humulene, while (*E*)-β-caryophyllene and α-humulene together have shown strong synergistic effects [50].

*Callicarpa nudiflora* essential oil is characterized by 62.1% monoterpenoid compounds, and this may be one of the reasons responsible for its potent AChE inhibitory activity. β-Pinene, caryophyllene oxide, and myrtenal have been reported to have synergistic effects among them [50].

### 2.3. Homology Modeling Study of AChE1 Enzyme

The physicochemical properties of AChE1 analyzed using the Protparam webserver are presented in Table 4.

On the other hand, various data on the secondary structures of AChE1, including alpha helix, extended strands and random coil, were predicted using SOPMA (Table 5).

According to the data analyzed in Table 4, the AChE1 enzyme contains 91 amino acids, and its molecular weight is 10,716.71 Da. The theoretical pI value was predicted to be 4.76, which suggests that this enzyme has a negative charge. According to Gupta et al. [54], the aliphatic index shows the relative volume occupied by aliphatic residues such as lysine, valine, leucine, and isoleucine. The low value predicted for this parameter (44.95) means that the enzyme cannot be stable in a wide range of temperature. The obtained results presented in Table 5 indicate that the random coil and alpha helix are the main components of the AChE1 enzyme (45.05% and 36.26%, respectively). The constructed model did not possess a beta sheet or turns in the secondary structure.

The modeled structure of AChE1 with a potential binding pocket was predicted using CASTp tool (Figure 2A). A model can be considered to be good when the QMEAN4 score varies between 0 and 1 [55]. In this study, the QMEAN4 score of AChE1 was 0.45, thus, it might be argued that the predicted model is reliable for the performance of further docking studies. In addition, a more negative value of QMEAN4 Z-score indicates the low quality of the constructed model compared with the template structure. The QMEAN4 Z-score of AChE1 model was recorded to have a value of −1.06, suggesting that its quality is significantly high compared to the template structure 6ARX_A (Figure 2B). The predicted structure of AChE1 was also superimposed on the template model using the TM-align tool [56], and the obtained RMSD value was 0.11 Å. The homology modeling structure was then validated using the SAVES webserver (https://saves.mbi.ucla.edu (accessed on: 20 July 2022). The obtained Ramachandran plot shows that 94.8% of the residues were located in the most favorable region, and only 5.2% of the residues were located in the allowed region (Figure 2C). No residues were found in the outlier region. The ERRAT program gives an overall quality factor of 100, suggesting that this model is highly reliable (Figure 2D).

For further analysis, *Ee*AChE and AChE1 enzyme models were superimposed to validate the accuracy of the homology modeling process. The structure validation between the *Ee*AChE and AChE1 enzyme models was executed using Chimera 1.13.1. Obtained results indicate these models matched with high identity (Figure 3); therefore, the *Ee*AChE model will be chosen as a representative structure for docking studies in the latter stage.

### 2.4. Molecular Docking Studies

The best docking conformation of galantamine inside human AChE (PDB ID: 4EY6) was superimposed with the native ligand. The Root Mean Square Deviation (RMSD) value obtained for the superimposition was 0.714 Å (Figure 4).

The binding free energies, ligand efficacy, and residues participating in the interaction of the studied compounds are tabulated in Table 6.

Lowest-energy docked poses of the studied compounds suggested by docking simulation are presented in Figure 5.

Autodock4 is a common docking tool with approximately 6000 citations since 2009 [57]. This program is an open-source package that can predict the binding affinity and dock pose of ligands toward a specific protein target [58]. It has been reported by Gohlke et al. that ligand partial charge calculated using the PM6 method has been shown to greatly increase the docking accuracy and cluster population of the most accurate docking [59]. In this study, the main components of the studied essential oils were studied using Autodock4 to allow a deeper insight into their mechanism of inhibition against the *Ee*AChE and AChE1 enzymes. Initially, galantamine was redocked towards the human AChE enzyme to validate the docking procedure. According to Gowthaman et al., when the RMSD of the dock pose of the co-crystallized ligand is less than 2.0 Å in relation to the native crystallographic pose, docking validation can be considered to be satisfactory [60]. Retrieving the dock pose of co-crystallized ligands, it was possible to validate the docking protocol.

According to the ranking criteria of Autodock4, the more negative the value of the docking score, the higher the binding affinity of the compound towards the targeted receptor [61,62]. The obtained dock score of galantamine was −12.76 kcal/mol; thus, any ligands whose docking energy was close to this threshold would be assumed to exhibit good binding affinity toward the targeted enzyme. As indicated in Table 6, compounds with low docking scores might be assumed to be potential *Ee*AChE inhibitors; these results show good agreement with the inhibition activities obtained from the enzymatic studies.

Binding orientation analysis of galantamine indicated that Glu202 and Ser203 are the key residues participating in forming H-bonds with the reference ligand. This interaction was further stabilized by the hydrophobic bond with Trp86, Gly121, Gly122, Tyr124, Phe295, Phe297, Tyr337, Phe338, His447 (Figure 5). It should be noted that Trp86, Phe297, Tyr337, Tyr341 and His447 have been reported to participate in constituting the active site of the *Ee*AChE enzyme [63,64].

Of all the docked compounds, limonene exhibited the most negative value of binding free energy (−9.78 kcal/mol) towards *Ee*AChE, suggesting that it binds to this enzyme with the highest binding affinity. Dock pose analysis with the targeted enzyme revealed that Phe297, Phe338 and Tyr341 were the key residues contributing significantly to achieving good docking scores.

The remaining components were ranked in the following order: β-pinene, (*E*)-β-caryophyllene, α-humulene, *trans*-carveol and caryophyllene oxide, with Autodock4 docking scores of −9.24; −8.97; −6.95; −6.35 and −5.00 kcal/mol, respectively. An array of hydrophobic interactions was observed, which were contributed by Trp86, Tyr337, His447 with β-pinene. Binding mode analysis of (*E*)-β-caryophyllene showed this compound shared a common non-polar interaction with essential residue Phe297 in comparison to galantamine. The docked pose was strengthened by additional binding toward Trp286. Regarding the remaining compounds, including α-humulene, *trans*-carveol, and caryophyllene oxide, despite of their docking conformation with important residues within the binding site of targeted enzyme, the high value of docking scores suggests they have lower potential to be considered as inhibitors for *Ee*AChE.

### 2.5. FPL Simulation

Although the docking protocol often produces suitable results when compared with the experimental results, this method does not consider the receptor dynamics and limiting the number trial position of ligands, which might ultimately result in inaccurate predictions. Thus, a more accurate and precise method could be employed to refine the docking observation. Among various available techniques, the fast pulling of ligand (FPL) technique is a very efficient method with low required CPU time consumption that is able to provide results with high accuracy and precision. In particular, a ligand is forced to travel from bound to unbound states via a harmonic external force (Figure 6). The physical details during the unbinding process reveal the binding affinity and mechanism of a ligand to the AChE enzyme.

The relative binding affinity of a ligand to the AChE enzyme was estimated using eight independent FPL calculations. The pulling force was recorded every 0.1 ps, and other metrics were monitored every 10 ps. All of the computed values were averaged over eight independent trajectories. The recorded pulling force and work are shown in Figure 7.

The maximum pulling force (F_max_), called the rupture force, and the recorded pulling work (W) were used as a criterion for evaluating the ligand affinity. As mentioned in the previous work [65], the pulling work is more appropriate than the rupture force, as it is directly associated with ligand-binding free energy via the isobaric–isothermal Jarzynski equality. The data obtained showed that the pulling forces continuously increased to maximum values before rapidly dropping to zero after the nonbonded contacts between the ligand and the protein were terminated after 400 ps to 700 ps.

### 2.6. Drug-Likeness Studies

As a follow-up to the obtained docking results, the ADMET properties of the “hit” compounds were then analyzed, including human intestinal absorption (HIA), blood–brain barrier (BBB), carcinogenicity, and tumorigenesis potential (Table 7).

It is estimated that a compound with logPS > −3 can be considered to penetrate the central nervous system [66]. The obtained data indicate that all studied compounds satisfy the basic criteria to be a potential pesticide or drug, with HIA values ranging from 94.09% to 96.46%, and the CNS values were determined to be within the range −1.847 to −2.637. It should be noted that caryophyllene oxide was also predicted to have tumorigenic effect, suggesting further caution be used when applied in treatment.

## 3. Conclusions

Essential oils from traditional Vietnamese medicinal plants were studied for their AChE inhibitory activity, with most of them showing good inhibitory activity. Species such as *C. candicans*, *C. erioclona*, *C. rubella*, *C. nudiflora*, *P. corymbosa*, *V. trifolia* subsp. *litoralis* and *V. trifolia* subsp. *trifolia* are widely distributed, and their essential oils may be considered to be renewable raw materials. Most of the essential oils in this study were characterized by an absolute predominance of sesquiterpenoids, and *C. erioclona*, *C. sinuata* and *C. petelotii* essential oils exhibited the strongest AChE inhibition. For all of the essential oil components studied, reasonably low docking scores were obtained, which is in good agreement with enzymatic studies showing them to be potential *Ee*AChE and AChE1 enzyme inhibitors. Current knowledge is insufficient to explain these cases, however, and further studies are needed on AChE inhibition by purified monoterpenoids and sesquiterpenoids and their synergistic and antagonistic effects.

## 4. Materials and Methods

### 4.1. Source of Essential Oil

Details of the isolation methods of essential oils are available in our previous articles [27,28,29]. The extraction yield of the essential oils is presented in Table 8.

### 4.2. Acetylcholinesterase (AChE) Inhibition Assay

Acetylcholinesterase (AChE) inhibitory activity of essential oil was determined according to the method described by Ellman and our previous study [67,68]. The stock solution was obtained by dissolving the essential oil in DMSO (Merck), which was then diluted with H_2_O (deionized distilled water) to obtain different experimental concentrations. Each solution mixture consisted of 140 μL of phosphate buffer solution (pH: 8), 20 μL of essential oil at concentrations of 500, 100, 20, and 4 μg/mL, and 20 μL of the enzyme AChE (0.25 IU/mL, Sigma-Aldrich, St. Louis, MO, USA). The reaction mixtures were transferred to the test wells of a 96-well microtiter plate and incubated at 25 °C for 15 min. Then, solutions of 10 μL dithiobisnitrobenzoic acid (DTNB, 2.5 mM, Sigma-Aldrich) and 10 μL acetylthiocholine iodide (ACTI, 2.5 mM, Sigma-Aldrich) were added to each of the test wells and incubation continued for 10 min at 25 °C. At the end of the experiment, the absorbance of each solution was measured at 405 nm. Galantamine was used as a positive control. The negative control well did not contain the test sample. Each test was carried out in triplicate.

### 4.3. Molecular Docking Studies

The major components in the essential oil of studied species were selected for the docking study. The three-dimensional structures of the studied compounds were prepared using MarvinSketch 19.27.0 and PyMOL version 1.3r1 [69]. Energy minimization of studied ligands were conducted using MM2 force field and quantum chemical calculations were performed using the PM6 semiempirical method implemented in Gaussian 09 [70]. Galantamine, a tertiary alkaloid acetylcholinesterase inhibitor, was used as the reference ligand.

The X-ray crystal structure of *Ee*AChE from Electrophorus electricus was retrieved from the Protein Data Bank (RCSB) with PDB ID: 1C2B [71]. The tertiary structure of acetylcholinesterase 1 (AChE1) from *Aedes aegypti* has not been well determined in previous studies; thus, the structure was constructed by comparative modeling using the SWISS-MODEL webserver (http://swissmodel.expasy.org (accessed on: 20 July 2022). The PDB entry 6ARX_A was selected as template structure for modeling AChE1 enzyme. The amino acid sequence of AChE1 enzyme was already determined and its information is publicity at UniProtKB, archieved under entry ID: UniProtKB-Q8MYC0. The Protparam webserver (https://web.expasy.org/protparam (accessed on: 20 July 2022) was used to calculate the physicochemical properties of the enzyme. To predict the secondary structure of the AChE1 enzyme, the SOPMA tool was used (https://npsa-prabi.ibcp.fr/cgi-bin/npsa_automat.pl?page=/NPSA/npsa_sopma.html (accessed on: 20 July 2022). The predicted 3D structure was validated using PROCHECK to evaluate backbone conformation based on Psi/Phi Ramachandran plot analysis. AutoDockTools (The Scripps Research Institute, CA, USA) [72] was employed to set up and performed docking calculation. To turn the protein molecule into a free receptor, the heteroatoms including water molecules were deleted. Then, the Kollman charges and solvation parameters were assigned, and Gasteiger charges were added to each atom.

The molecular docking study uses AutoDock4 with Lamarckian genetic algorithm (LGA) to search for the optimum dock pose together with the scoring function to calculate the binding affinity. The binding sites of *Ee*AChE and AChE1 were enclosed in a box with a number of grid points in the x × y × z directions of 50 × 50 × 50, and a grid spacing of 0.375 Å. Initially, AutoGrid was run to generate the grid map of various atoms of the ligands and receptor. After the completion of the grid map, AutoDock was run by using autodock parameters as follows: GA population size, 300; maximum number of energy evaluations, 25,000,000; and number of generations, 27,000. A maximum of 50 conformers were considered for each molecule, and the root-mean-square (RMS) cluster tolerance was set to 2.0 Å in each run. Ligand conformation with the lowest free energy of binding, chosen from the most favored cluster, was selected for the further analysis. The outputs of AutoDock modeling studies were analyzed using PyMOL version 1.3r1 [69] and Discovery Studio Visualizer (San Diego, CA, USA) [73].

As no experimental structure of *Ee*AChE from *Electrophorus electricus* complexed with an inhibitor is available on the RCSB archive, galantamine was redocked over itself in the crystal structure of human AChE complexed with galantamine (PDB ID: 4EY6) [74] to validate the accuracy of the docking protocol.

### 4.4. Drug-Likeness Studies

Open bioactivity prediction online server Molinspiration (https://www.molinspiration.com (accessed on: 20 July 2022) and pkCSM (http://biosig.unimelb.edu.au/pkcsm/prediction (accessed on: 20 July 2022) were used to evaluate drug-like properties. OSIRIS Property Explorer (http://www.organicchemis try.org/prog/peo (accessed on: 20 July 2022) was used to predict side effects, such as mutagenic and tumorigenic effects.

### 4.5. Binding Affinity Calculation

To follow up the Autodock 4.2 docking, the Molecular Mechanics/Generalized Born and Surface Area (MM-GBSA) method was used to predict binding free energy using Schrödinger software 2020-2 [74]. This energy is determined by the difference between the complex and the specific energy of the protein and ligand (Equation (1)). This total energy has its heat of reaction calculated by the summary of the solvation free energy (∆Gsol), the gas-phase interaction energy (∆Egas) and the entropy terms (∆S) (Equation (2)). At constant pressure, the heat of the reaction is approximately equal to the change of internal energy, and thus ∆Eint is neglected (Equation (3)). The remaining form of energy includes all intermolecular interactions, for example electrostatic interactions, protein–ligand vdW interactions, ligand desolation, internal strain energies with OPLS2005 force field. The solvation free energy (∆Gsol) consists of non-polar (∆Gsurf) and polar (∆GGB) energy forms and their solvation energy is calculated by sum of the solvent-accessible surface area and GB model (Equation (4)). In general, the binding free energy (∆Gbind) is determined by the sum of solvation free energy and gas-phase interaction energy. At the same time, the entropy term is neglected in the calculation for relative free binding energies [75,76].
ΔG_bind_ = G_complex_ − G_receptor_ − G_ligand_(1)
ΔG_bind_ = ΔH − TΔS ≈ ΔE_gas_ + ΔG_sol_ − TΔS(2)
ΔE_gas_ = ΔE_int_ + ΔE_ELE_ + ΔE_VDW_(3)
ΔG_sol_ = ΔG_GB_ + ΔG_surf_(4)

In summary, the MM-GBSA calculations are not in agreement with experimental binding affinities, but they show the tendency to bind and the reasonable correlation with the experiment values, and the more negative value of MM-GBSA indicates more potent approximate free energies of binding.

### 4.6. Fast Pulling Ligand Simulation

The structure of the complex was obtained via the molecular docking method. Caver 2.1 [77] was employed to evaluate the disassociate direction of a ligand as suggested by previous works [78]. The complex was then aligned to be the disassociate pathway oriented to the Z-axis. The complex was inserted into a periodic boundary condition rectangular box (7.26 × 9.23 × 12.35 nm) consisting of an enzyme AChE, a ligand, 22,761 water molecules, and 7 counterbalanced ions (Na^+^). In particular, the protein was parameterized via the AMBER99SB-ILDN force field [79], and the ligand was represented using general Amber force field [80]. The water molecule was topologized using TIP3P water model [81].

GROMACS version 2016 was employed to carry out the molecular dynamics (MD) simulations. The simulation was performed according to the following four steps: energic minimization, NVT, NPT, and steered MD (SMD) simulations. Specifically, the non-covalent pair was affected within a range of 0.9 nm, and the pair list was updated every 5 fs. The particle mesh Ewald method [82] was employed to mimic the electrostatic interaction with cut-off of 0.9 nm. The van der Waals interaction was affected in a range of 0.9 nm. Both of NVT and NPT simulations were carried out with a length of 100 ps at 300 K. During simulations, the AChE C_α_ atoms were restrained by using a slight harmonic force. The last snapshot of NPT simulations was used as initial structure of SMD simulations. In this scheme, a harmonic external force with a cantilever spring constant of k = 600 kJ/mol/nm^2^ and pulling speed v = 0.005 nm/ps was put on the ligand center of mass along the Z-direction. The ligand was disassociated from the binding site of AChE enzyme over the SMD simulations. The data were recorded every 0.1 ps during SMD simulations. The calculations were repeated independently eight times with the same initial conformation to guarantee sufficient sampling.

### 4.7. Data Analysis

Inhibitory data were analyzed by log-probit analysis [83] to acquire IC_50_ value as well as 95% confidence limits using Minitab^®^ version 19.2020.1 (Minitab, LLC, State College, PA, USA).

## Figures and Tables

**Figure 1 molecules-27-07092-f001:**
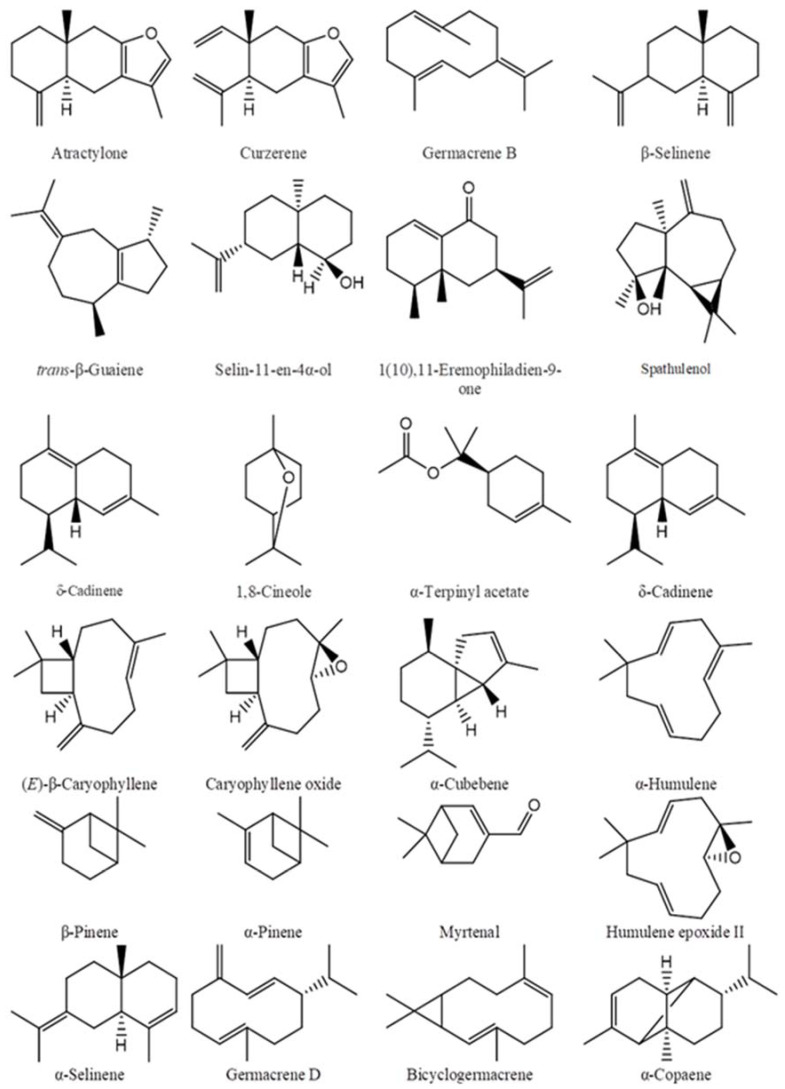
The main chemical components of essential oils.

**Figure 2 molecules-27-07092-f002:**
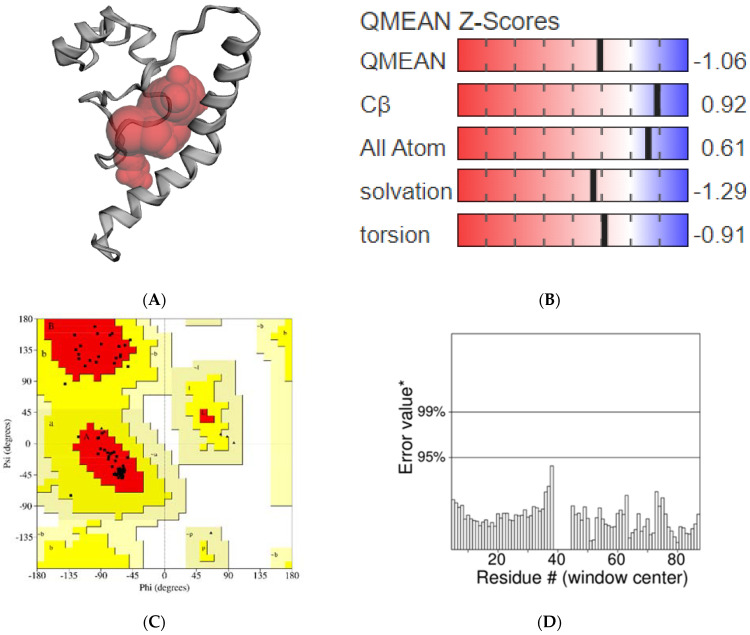
Validation of the predicted structure of AChE1. (**A**) Binding pocket predicted by CASTp; (**B**) QMEAN Z−Scrores value; (**C**) Ramachandran plot; (**D**) ERRAT plot. Note: Non-glycine and non-proline residues are shown as squares; Glycine residues are shown as triangles; [A, B, L]: Residues in most favoured regions; [a, b, l, p]: Residues in additional allowed regions; [~a, ~b, ~l, ~p]: Residues in generously allowed regions; Error value*:On the error axis, two lines are drawn to indicate the confidence with which it is possible to reject regions that exceed that error value; Residue #: Indate the residue number in protein model.

**Figure 3 molecules-27-07092-f003:**
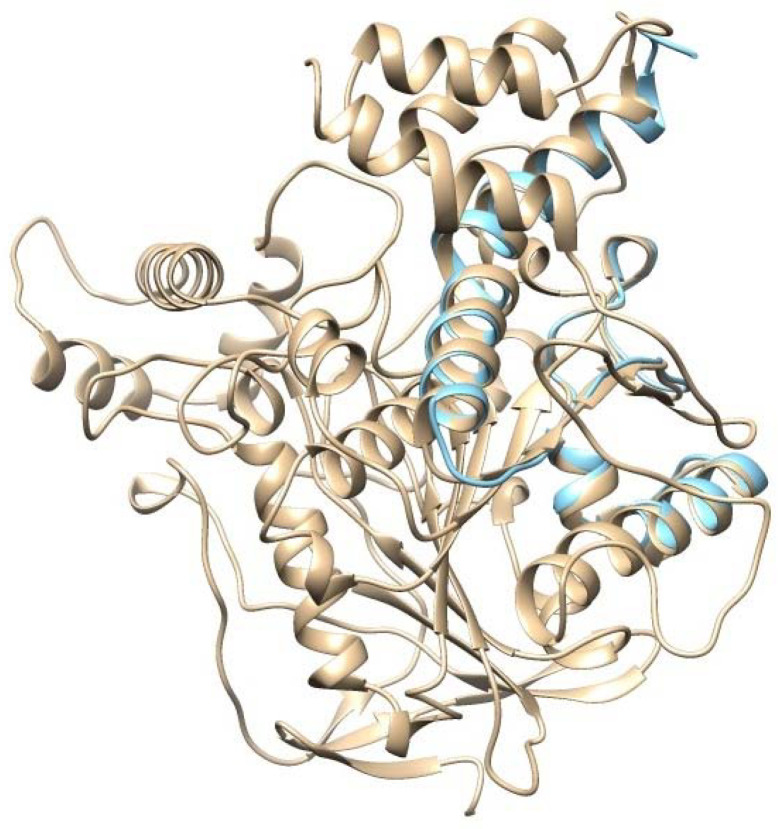
Overlay of *Ee*AChE and AChE1 models produced using Chimera 1.13.1; the *Ee*AChE model is presented in apricot color; the AChE1 model is presented in cyan color.

**Figure 4 molecules-27-07092-f004:**
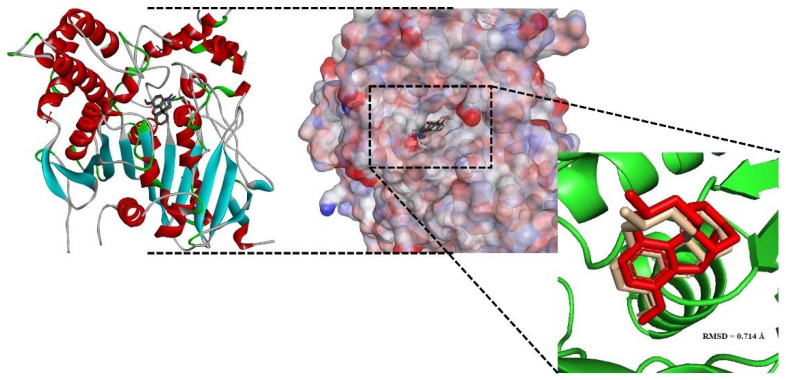
Dock pose overlay of crystallographic ligands (red) with the calculated shape (gray).

**Figure 5 molecules-27-07092-f005:**
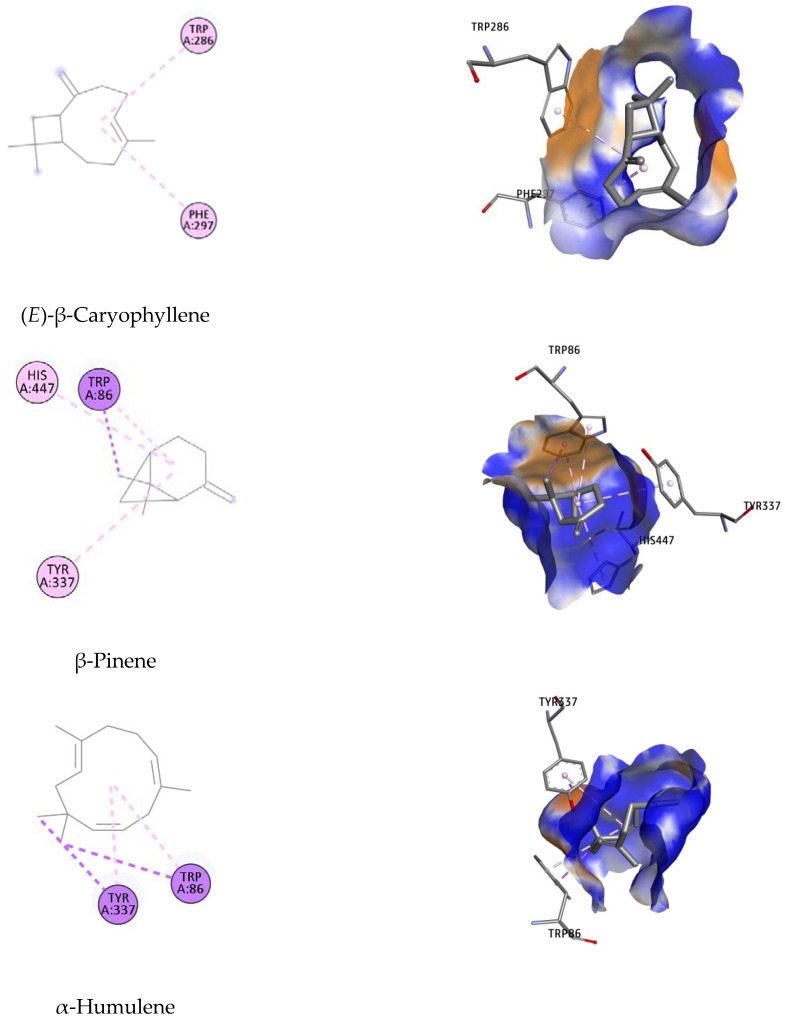
Binding orientation of potential inhibitors at the binding site of the AChE enzyme, suggested on the basis of molecular docking studies.

**Figure 6 molecules-27-07092-f006:**
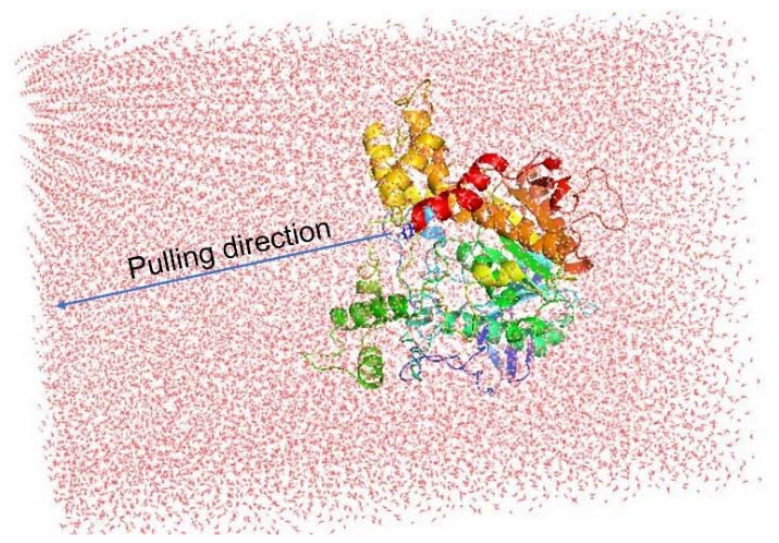
Computational modeling of FPL calculations. The pulling pathway is aligned along the Z−axis.

**Figure 7 molecules-27-07092-f007:**
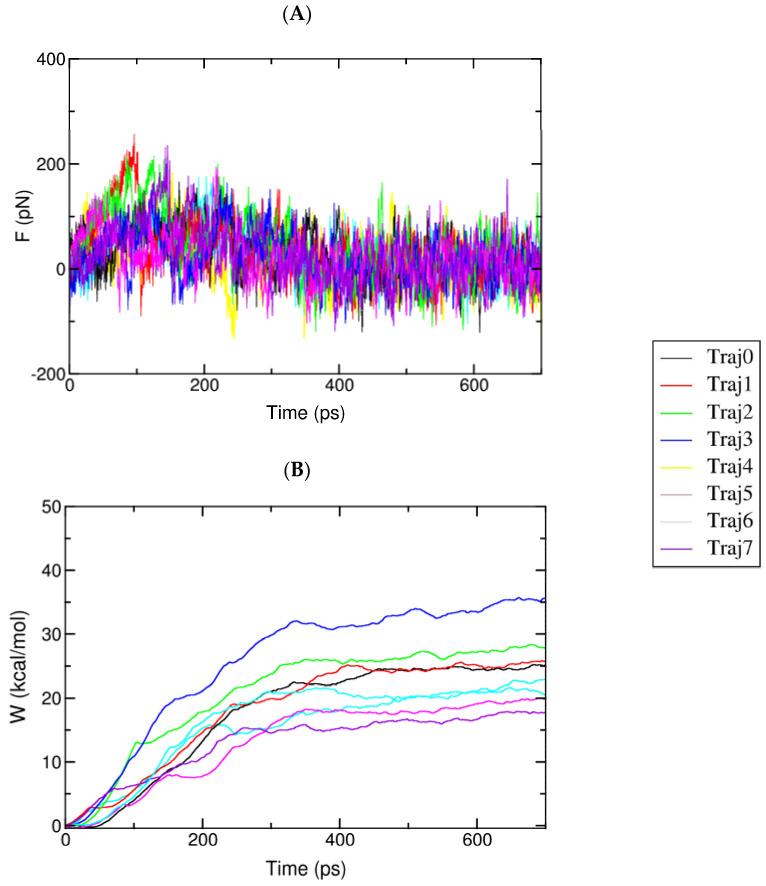
(**A**) The mean force of complexes over eight independent SMD trajectories. The data were monitored every 0.1 ps; (**B**) The mean work of complexes during eight independent SMD trajectories.

**Table 1 molecules-27-07092-t001:** Major components of the leaf essential oils of *Callicarpa, Premna, Vitex* and *Karomia* species from Vietnam.

Essential Oil	Major Components of the Essential Oils	Classification
*Callicarpa candicans* (Nghe An) [27]	Atractylone (37.7%), (*E*)-β-caryophyllene (19.0%), β-selinene (6.2%).	Ses: 95% (hydro: 49%; oxy: 46%)
*Callicarpa candicans* (Da Nang) [27]	Atractylone (42.4%), (*E*)-β-caryophyllene (15.3%), curzerene (5.3%), germacrene B (5.1%), β-selinene (4.5%).	Ses: 92.2% (hydro: 40.7%; oxy: 51.5%)
*Callicarpa candicans* (Quang Nam) [27]	Caryophyllene oxide (13.4%), (*E*)-β-caryophyllene (7.1%), β-selinene (5.7%).	Ses: 52.5% (hydro: 22.6%; oxy: 29.9%) compared with 59.7% of the identified compounds.
*Callicarpa longifolia* [27]	*trans*-β-Guaiene (22.2%), (*E*)-β-caryophyllene (11.8%), selin-11-en-4α-ol (8.0%), 1(10),11-eremophiladien-9-one (6.7%).	Ses: 92.4% (hydro: 63.0%; oxy: 29.4%).
*Callicarpa sinuate* [27]	α-Humulene (24.8%), α-copaene (12.6%), humulene epoxide II (6.7%), spathulenol (5.9%).	Ses: 91.6% (hydro: 67.5%; oxy: 24.1%).
*Callicarpa petelotii* [27]	α-Humulene (53.8%), α-selinene (12.8%), humulene epoxide II (8.1%).	Ses: 91.0% (hydro: 74.7%; oxy: 16.3%).
*Callicarpa rubella* (Tay Giang) [27]	(*E*)-Caryophyllene (18.0%), α-cubebene (17.4%).	Ses: 87.5% (hydro: 78.5%; oxy: 9.0%).
*Callicarpa nudiflora* [27]	β-Pinene (34.2%), caryophyllene oxide (20.1%), α-pinene (8.1%), myrtenal (6.8%).	Mono: 62.1% (hydro: 47.5%; oxy: 14.6%); ses: 32.8% (hydro: 5.7%, oxy: 27.1%).
*Callicarpa erioclona* *	Atractylone (34.6%), (*E*)-β-caryophyllene (11.1%), caryophyllene oxide (5.9%), β-selinene (5.1%).	Ses: 75.4% (hydro: 32.1%; oxy: 43.3%) compared with 79.6% of the identified compounds.
*Callicarpa macrophylla* (Pu Mat) *	(*E*)-Caryophyllene (25.2%), caryophyllene oxide (6.4%).	Ses: 73.9% (hydro: 62.0%; oxy: 11.9%).
*Prmena cambodiana* [28]	α-Copaene (23.3%), (*E*)-caryophyllene (12.8%), α-gurjunene (11.3%), δ-cadinene (5.5%).	Ses: 88.4% (hydro: 76.1%; oxy: 12.3%).
*Prmena corymbosa* [28]	*allo*-Aromadendrene (39.7%), (*E*)-caryophyllene (13.3%), α-copaene (8.1%).	Ses: 93.6% (hydro: 85.4%; oxy: 8.2%).
*Vitex ajugifolia* [29]	α-Copaene (17.0%), (*E*)-β-caryophyllene (11.7%), α-humulene (9.6%), spathulenol (8.7%).	Ses: 97.8% (hydro: 68.2%; oxy: 29.6%).
*Vitex trifolia* subsp. *litoralis* [29]	α-Pinene (18.7%), sabinene (15.2%), 1,8-cineole (14.5%), α-terpinyl acetate (12.7%).	Mono: 76.0% (hydro: 44.9%; oxy: 31.1%); ses: 1.6% (hydro: 1.1%, oxy: 0.5%).
*Vitex pinnata* [29]	(*E*)-β-Caryophyllene (32.7%), germacrene D (17.1%), bicyclogermacrene (11.1%).	Ses: 95.8% (hydro: 87.1%; oxy: 8.7%).
*Vitex trifolia* subsp. *trifolia* [29]	Sabinene (19.4%), 1,8-cineole (15.7%), (*E*)-β-caryophyllene (14.5%), α-pinene (11.7%), α-terpinyl acetate (8.3%).	Mono: 38.4% (hydro: 17.6%; oxy: 20.8%); ses: 39.2% (hydro: 31.5%, oxy: 7.7%).
*Karomia fragrans* *	(*E*)-Caryophyllene (26.5%), caryophyllene oxide (10.5%), α-humulene (10.1%), p-cymene (7.5%), δ-cadinene (5.2%).	Ses: 73.1% (hydro: 51.5%; oxy: 21.6%).

*: These are/have been published elsewhere, and are included here for comparison purposes. Ses: Sesquiterpenoids. Mono: Monoterpenoids. Hydro: hydrocarbons. Oxy: Oxygenated. Nghe An, Da Nang, Quang Nam, Tay Giang, Pu Mat: plant material collected in Nghe An, Da Nang, Quang Nam, Tay Giang, Pu Mat.

**Table 2 molecules-27-07092-t002:** Acetylcholinesterase inhibitory activity of essential oils of *Callicarpa, Premna*, *Vitex* and *Karomia* species from Vietnam.

Concentration (µg/mL)	*C. candicans* (Nghe An)	*C. candicans* (Da Nang)	*C. candicans* (Quang Nam)	*C. longifolia*		*C. sinuata*		*C. petelotii*	*C. rubella* (Tay Giang)	*C. nudiflora*
500	87.88	86.37	88.34	82.72		82.13		90.66	89.78	89.78
100	59.57	66.53	61.00	52.77		66.71		62.70	52.24	69.35
20	32.61	37.46	34.05	19.55		50.91		47.36	25.60	25.26
4	5.28	10.42	10.27	8.06		15.47		21.03	7.12	3.67
IC_50_	58.38 ± 2.95	45.67 ± 1.84	52.88 ± 3.19	105.16 ± 5.61		34.15 ± 1.35		32.11 ± 3.85	89.38 ± 4.05	54.69 ± 3.05
**Concentration (µg/mL)**	** *C. erioclona* **	** *C. macrophylla* ** **(Pu Mat)**	** *P. cambodiana* **	** *P. corymbosa* **	** *V. ajugifolia* **	***V. trifolia* subsp. *litoralis***	** *V. pinnata* **	***V. trifolia* subsp. *trifolia***		** *K. fragrans* **
500	100.00	75.00	81.83	97.55	82.58	103.82	83.79	82.02		75.67
100	74.06	33.60	42.30	59.14	65.02	49.53	44.37	57.21		40.66
20	44.70	6.72	17.09	24.67	38.06	3.79	23.22	22.92		6.43
4	26.77	−1.56	8.61	11.54	7.55	−4.77	17.59	−1.19		−0.41
IC_50_	28.71 ± 3.24	221.85 ± 15.32	157.06 ± 4.14	73.35 ± 4.61	50.93 ± 4.81	120.32 ± 16.64	144.33 ± 16.94	81.34 ± 3.57		187.91 ± 14.09

**Table 3 molecules-27-07092-t003:** Acetylcholinesterase inhibitory activity of main components of the essential oils of *Callicarpa, Premna, Vitex*, and *Karomia* species from Vietnam ^a^.

Concentration (µg/mL)	(*E*)-β-Caryophyllene	β-Pinene	α-Humulene	Limonene	Caryophyllene Oxide	*trans*-Carveol	Galantamine ^b^
500	79.05	86.54	72.83	90.82	74.00	76.90	---
100	52.89	58.86	44.27	67.73	16.95	55.34	---
20	26.28	28.88	25.10	27.04	0.49	16.79	---
4	6.09	20.92	19.37	18.17	−2.11	1.19	---
IC_50_ (μg/mL)	89.10 ± 6.10	71.45 ± 5.77	160.48 ± 13.48	53.16 ± 4.08	320.16 ± 13.47	102.88 ± 7.84	1.78 ± 0.13b
IC_50_ (μM)	436.0 ± 29.9	524.5 ± 42.4	785.3 ± 66.0	390.2 ± 30.0	1453 ± 61	675.8 ± 51.5	6.19 ± 0.45

^a^ Data are presented as IC_50_ values ± standard deviations obtained graphically from four independent experiments carried out in triplicate. ^b^ Galantamine was tested at concentrations of 10, 2, 0.4, and 0.08 µg/mL.

**Table 4 molecules-27-07092-t004:** Physicochemical properties of AChE1 analyzed using the Protparam webserver.

Parameters	
Number of amino acids	91
Molecular weight	10,716.71
Theoretical pI	4.76
Aliphatic index	44.95

**Table 5 molecules-27-07092-t005:** Percentage of secondary structures of AChE1 predicted using SOPMA.

Secondary Structure	Number of Amino Acids	Percentage (%)
Alpha helix	33	36.26
310 helix	0	0.00
Pi helix	0	0.00
Beta bridge	0	0.00
Extended strand	13	14.29
Beta turn	4	4.40
Bend region	0	0.00
Random coil	41	45.05
Ambiguous states	0	0.00
Other states	0	0.00

**Table 6 molecules-27-07092-t006:** Dock score and MM-GBSA estimation and residue interactions of the studied compounds against AChE enzyme.

Ligand	Dock Score (kcal/mol)	MM-GBSA (kcal/mol)	Interacting Residues
(*E*)-β-Caryophyllene	−8.79	−105.41	Trp286, Phe297
β-Pinene	−9.24	−116.32	Trp86, Tyr337, His447
α-Humulene	−6.95	−43.19	Trp86, Tyr337
Limonene	−9.78	−145.78	Phe297, Phe338, Tyr341
Caryophyllene oxide	−5.00	−40.56	Phe338, Tyr341
*trans*-Carveol	−6.35	−46.22	Ile294, Phe295, Phe297, Tyr337, Phe338, Tyr34
Galantamine	−12.76	−154.63	Trp86, Gly121, Gly122, Tyr124, Glu202, Ser203, Phe295, Phe297, Tyr337, Phe338, His447

**Table 7 molecules-27-07092-t007:** Pharmacokinetic properties of studied compounds.

Compound	HIA (%) ^a^	CNS (logPS) ^b^	Mutagenic	Tumorigenic
(*E*)-β-Caryophyllene	94.09	−2.139	NO	NO
β-Pinene	95.43	−1.847	NO	NO
α-Humulene	94.43	−2.542	NO	NO
Limonene	95.40	−2.356	NO	NO
Caryophyllene oxide	95.88	−2.518	NO	YES
*trans*-Carveol	94.69	−2.637	NO	NO
Galantamine	96.46	−2.559	NO	NO

^a^ HIA: human intestinal absorption; ^b^ CNS: central nervous system.

**Table 8 molecules-27-07092-t008:** Extraction yield of essential oils.

Essential Oil	Yield (% *v*/*w*)	Part
*Callicarpa candica*ns (Nghe An)	0.15	Leaves
*Callicarpa candicans* (Da Nang)	0.17	Leaves
*Callicarpa candicans* (Quang Nam)	0.18	Leaves
*Callicarpa longifolia*	0.13	Leaves
*Callicarpa sinuata*	0.14	Leaves
*Callicarpa petelotii*	0.22	Leaves
*Callicarpa rubella* (Tay Giang)	0.12	Leaves
*Callicarpa nudiflora*	0.14	Leaves
*Callicarpa erioclona*	0.19	Leaves
*Callicarpa macrophylla* (Pu Mat)	0.24	Leaves
*Prmena cambodiana*	0.14	Leaves
*Prmena corymbosa*	0.25	Leaves
*Vitex ajugifolia*	0.09	Leaves
*Vitex trifolia* subsp. *litoralis*	0.12	Leaves
*Vitex pinnata*	0.14	Leaves
*Vitex trifolia* subsp. *trifolia*	0.12	Leaves
*Karomia fragrans*	0.12	Leaves

## Data Availability

All data are available from the corresponding author (V.T.H.) upon reasonable request.

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
