# Peer review of "Acetylcholinesterase Inhibitory Activities of Essential Oils from Vietnamese Traditional Medicinal Plants"

_molecules, 2022, doi:10.3390/molecules27207092_

Round 1

Reviewer 1 Report

The article by Hung et al. "Anticholinesterase Inhibitory Activities of Essential Oils from Vietnamese Traditional Medicinal Plants" describes in-vitro and computational studies of essential oils against AD. Well the article has some interesting results but at this condition, it is not suited for the molecules journal, But after careful Major revision, the article can be considered. I here by recommend some more work to this article which at least make it strong scientifically.

1. First of all, only molecule docking is not enough for these compounds. I strongly recommend to at least do the Molecular docking simulations of any one good compound for 50/100ns. and the authors already mentioned in the abstract that Molecular docking simulations were performed but I didn't see MDS, and on line251-252 the authors mentioned RMSD in the figure.2 but I cannot see and RMSD file in fig.2.

2. Furthermore, I also recommend doing the MM.GBSA energy calculation.

3. The authors should also do additional ADMET data in their article. As ADMET is one of the most important portions for any biomolecule. i recommend here one article. The authors can follow this article and can add in-silico data to their article so the article can be more strong Molecules 202227(3), 917;

4. AFter adding these data also update your abstract.

Reviewer 2 Report

The MS deals with the anticholinesterase activity of some essential oils from Vietnamese medicinal plants. IC50 values were determined and compared to each other. The MS is well-written, easy to follow but needs some improvement.

Detailed comments:

Title: Please change anticholinesterase to acetylcholinesterase.

Line 95 and title of Table 1, 2 and 3: The Vitex species are not mentioned in the text, although results considering this genus are shown in the Tables.

Table 1: What does it mean Nghe An, Da Nang and Quang N after the species name of Callicarpa candicans? Are they cultivars or the place of origin?

Line 132-133: It is hard to make comparison among the enzyme inhibitory activities because the used concentrations and also the degree of inhibition are very different. If you ad concentrations causing the same degree of inhibition, like in your tables, where IC50 is added, the interpretation of the data is more easier.

Line 329: Origin of the used AChE?  

Round 2

Reviewer 1 Report

The authors addressed all my questions very clearly.I recommend for its publication.